

# A Retrospective Streamflow Ensemble Forecast for an Extreme Hydrologic Event: a Case Study of Hurricane Irene and on the Hudson River basin

F. Saleh*, V. Ramaswamy, N. Georgas, A. F. Blumberg and J. Pullen

5   Stevens Institute of Technology, Davidson Laboratory, Department of Civil, Environmental and Ocean Engineering, Hoboken NJ, 07030, USA

*Correspondence to*: F. Saleh (fsaleh@stevens.edu)

**Abstract.** This paper investigates the uncertainties in hourly streamflow ensemble forecasts for an extreme hydrological event using a hydrological model forced with short-range ensemble weather prediction models.

10   A state-of-the art, automated, short-term hydrologic prediction framework was implemented using GIS and a regional scale hydrological model (HEC-HMS). The hydrologic framework was applied to the Hudson River Basin, USA (~36,000 km²) using gridded precipitation data from the National Centers for Environmental Prediction (NCEP) North American Regional Reanalysis (NARR) and was validated against streamflow observations from the United States Geologic Survey (USGS). Finally, 21 precipitation datasets reforecast by the 21 ensemble members of the latest Global Ensemble Forecast System (GEFS/R) were used to generate a retrospective streamflow ensemble forecast for an extreme hydrological event, Hurricane Irene. The work shows that ensemble stream discharge forecasts provide improved predictions and useful information about associated uncertainties, thus improving the assessment of risks when compared with deterministic forecasts. The uncertainties in weather inputs may result in false warnings and missed river flooding events, reducing the potential to effectively mitigate flood damage. The findings demonstrate how errors in the ensemble median streamflow forecast and time of peak, as well as the ensemble spread (uncertainty) are reduced 48 hours pre-event utilizing the ensemble framework. The methodology and implications of this work benefit efforts of short-term streamflow forecasts at regional scales, notably regarding the peak timing of an extreme hydrologic event when combined with a flood threshold exceedance diagram.

Although the modelling framework was implemented on the Hudson River basin, it is flexible and re-locatable in other parts of the world.

25   **Keywords**

Streamflow ensemble forecasts; Hudson River Basin, NARR, GEFS, Extreme hydrological event, Hurricane Irene, Weather models, hydrological ensemble skill assessment, uncertainty



# 1 Introduction

Riverine floods are known to adversely impact affected communities by causing casualties, inflicting damage to physical property, temporarily disrupting social and economic activities, and forcing a community to take emergency measures (IFRC, 2013). In the United States, for example, floods are recognized as the main natural disaster with $7.96 Billion in flood-related damages/year and 82 fatalities/year, averaged over the past 30 years (NWS, 2014). It is reported that 78% of emergencies are weather related (Weaver et al., 2014; Hoss and Fischbeck, 2016).

The increase in global averaged temperatures enhanced the potential for severe to extreme weather events (Becker and Grunewald, 2003; WMO, 2003). As the world warms, northern regions and mountainous areas are experiencing more precipitation falling as rain rather than snow, with a pronounced increase in precipitation being observed in the area of eastern North America (Karl, 2009). The special report on "Managing the Risks of Extreme Events and Disasters to Advance Climate Change Adaptation" of the Intergovernmental Panel on Climate Change (IPCC), critically assessed recent scientific literature on climate change and the impacts from extreme events. They reported that increased frequency and intensity of rainfall, based on climate models, substantially contributed in local flooding (Kundzewicz et al., 2014). Studies addressing flood damage in the United States show that the impact of floods to a community has increased over time as a result of both climate factors and societal factors: increased damage is associated with increased precipitation and with increasing population and urban development (Pielke Jr, 2000; Changnon et al., 2001). Furthermore, studies report that associated monetary damages from flooding are also likely to go up in the 21$^{st}$ century and beyond (Milly et al., 2002; Allamano et al., 2009; Pall et al., 2011)..
The rise in the number of extreme weather events in recent years has spurred the need to better predict floods and mitigate flood damage. Such advanced warnings are not only important for economic losses but can mean the difference between life and death (NWS, 2012) . Studies suggest that as little as one hour of lead-time can result in a ten-percent reduction in flood damages if forecast information is communicated in a timely manner (McEnery et al., 2005). Flood modelling and prediction has greatly advanced in recent years with the advent of geographic information systems (GIS), high-resolution digital elevation models (DEMs), distributed hydrologic and weather models and better delivery systems on the internet (McEnery et al., 2005). However, despite the advancement in hydrological prediction systems, such systems remain plagued by uncertainty from numerical weather prediction models (Clark and Hay, 2004) and, hydrologic model and structure parameters (Krzysztofowicz, 2001a; Gupta, 2005).

In terms of atmospheric forcing, the main source of uncertainty in streamflow forecasts arises from precipitation forecast errors which include errors arising from parameterizations of physical processes in atmospheric models, resolution and initial conditions (Krzysztofowicz, 2001a; Bartholmes and Todini, 2005; Cuo et al., 2011). In this context, ensemble hydrological forecasts using every ensemble are appealing to account for the uncertainty in numerical weather prediction (NWP) model forecasts (Buizza et al., 1999; Krzysztofowicz, 2001b; Bowler et al., 2008; Hamill et al., 2008; Cloke and Pappenberger, 2009).



Recent studies show the promise of adopting streamflow ensemble forecast techniques due to advantages over deterministic forecasts (Habets et al., 2004; Younis et al., 2008; Boucher et al., 2011; Schellekens et al., 2011; Verkade and Werner, 2011; Alfieri et al., 2013) as well as a way of accounting for uncertainties in hydrological forecasting (Chen and Yu, 2007; Demeritt et al., 2007; Davolio et al., 2008; Pappenberger et al., 2008; Reggiani and Weerts, 2008; Cloke and Pappenberger, 2009; Bao et al., 2011; Bogner and Pappenberger, 2011; Cuo et al., 2011; Schellekens et al., 2011; Alfieri et al., 2012; Amengual et al., 2015). Other advantages include the ability to distinguish between an extreme event forecast that is more or less likely to occur within the model's forecast horizon (Buizza, 2008; Golding, 2009) and better decision making with respect to operational hydrological concerns (Ramos et al., 2007; McCollor and Stull, 2008; Boucher et al., 2012). Furthermore, ensemble-based streamflow forecasts tend to be more consistent between successive forecasts (Pappenberger et al., 2011). Fan et al. (2014) showed benefits in the use of ensembles, particularly for reservoir inflows on flooding events, and in comparison to the deterministic values given by the control member of the ensemble and by the ensemble mean. Komma et al. (2007) found that for longer lead forecast times, the variability of the precipitation ensemble is amplified as it propagates through the catchment system as a result of non-linear catchment response. Also, the ensemble spread was found to be a useful indicator to assess potential forecast errors for lead time greater than 12 hours. Several techniques have been devised to account for uncertainty in hydrological forecast systems. Krzysztofowicz (2001a) implemented a method of combining the hydrological uncertainties with uncertainties of the precipitation forecasts using a Bayesian Forecasting System (BFS) which decomposes the total uncertainty about the river stage into precipitation uncertainty and hydrologic uncertainty, which are quantified independently and then integrated into a predictive distribution of the river stage. Montanari and Grossi (2008) indirectly related the forecast error to the sources of uncertainty in the forecasting procedure through a probabilistic link with the current forecast issued by the hydrologic model, the past forecast error, and the past rainfall. Olsson and Lindström (2008) performed analysis on separating the contributions of the precipitation forecast errors and the hydrological simulation errors. Weerts et al. (2011) used quantile regressions to assess the relationship between the hydrological forecast and the associated forecast error. Renard et al. (2010) addressed the total predictive uncertainty and separated it into input and structural components under different inference scenarios. They highlighted the inherent limitations of inferring inaccurate hydrologic models using rainfall-runoff data with large uncertainties. Brown (2015) quantified the total uncertainty in future streamflow as a combination of the meteorological forcing uncertainties and the hydrologic modeling uncertainties. He implemented a Meteorological Ensemble Forecast Processor (MEFP) to quantify the meteorological uncertainties and correct biases in the forcing inputs to the streamflow forecasts modeling.

In spite of the advancements and advantages in streamflow ensemble forecasts reported in the literature there are a number of key scientific questions that need better understanding. These include: how the meteorological forecast uncertainty reflects in the ensemble streamflow forecast; how does the degree of spread and hydrological response correlate with the lead time of the forecast and the scale of application; and how effective is streamflow ensemble forecasting during an extreme hydrological event. The present work investigates these questions by retrospectively forecasting streamflow of an extreme event (Hurricane Irene) in the Hudson River Basin, USA. Hurricane Irene had strong hydrological effects from high moisture content that



brought very heavy rainfall rates to the US East Coast including the Hudson River basin (Coch, 2012) (Figure 1). The total estimated damage from Hurricane Irene was ~$15.8 billion. This includes about $7.2 billion from inland flooding and storm surges (Avila and Cangialosi, 2011).

In this paper we first describe the case study area and context. We then summarise the main datasets that were used to

5    implement the hydrologic framework. Subsequently, we provide a detailed quantitative analysis and discussion of the uncertainties in the streamflow forecasts associated with the forcing from weather models and demonstrate how uncertainties in streamflow forecast median, time of peak and spread are reduced approaching a given event.

## 2 Materials and Methods

### 2.1 Study area and context

10   The study encompasses the Hudson River Basin (USA) which originates from the Adirondack Mountains of Upstate New York and drains into the Atlantic Ocean (Figure 2). The drainage area of the basin is approximately 36,000 km$^2$, covering 25% of New York State and other portions of the States of New Jersey, Connecticut, Massachusetts and Vermont. The basin is considered one of the largest drainage areas in the eastern seaboard of the United States. According to a national water quality assessment study conducted by the United States Geological Survey (USGS), nearly 60% of the water supplied in the basin is

15   for commercial or industrial use. Several reservoirs within the Hudson River basin contribute to the New York City water-supply system, which supplies water to about 8 million people.

In 2011, Hurricane Irene caused severe damage and widespread destruction that affected the east coast of the United States. The storm made landfall as a strong tropical storm at Little Egg Inlet in New Jersey on August 28, 2011 (Figure 1). The total precipitation accumulation from Hurricane Irene during August 27–30, 2011 was more than 300 mm in certain areas of the

20   Hudson River basin (Figure 4). It inundated streams throughout New Jersey resulting in peak stream flows exceeding the 100-year recurrence interval at many stream gages and causing heavy property and road damage. For instance, the Passaic and Hackensack River Basins in Northern New Jersey just south of the Hudson River Basin witnessed new record peaks at a number of streamflow-gauging stations.

President Obama issued a Major Disaster Declaration for counties in New York and New Jersey impacted by Hurricane Irene.

25   In total, 38 counties across New York State were impacted with an estimated $1.5 billion dollars in FEMA public assistance costs and 10 deaths (FEMA, 2011) . In New Jersey, the property damage was estimated to be $1 billion. In addition to high monetary damages, millions of people across the State were evacuated and seven deaths were reported (Watson et al., 2013; NJOEM, 2014).

### 2.2 Modeling framework description

30   The operational framework diagram is shown in Figure 3 and the datasets used in constructing the Hudson River basin regional scale hydrological model are depicted in Figure 2. The framework was validated using the National Centers for Environmental



Prediction (NCEP) North American Regional Reanalysis (NARR) precipitation data (Mesinger et al., 2006). A retrospective forecast of Hurricane Irene utilizing the 21 ensemble members from NOAA's Global Ensemble Forecast System Reforecast (GEFS/R) was then used (Hamill et al., 2015; Zhou and Zhu, 2016).

Apart from this work, the framework is currently operational and fully automated on the *Pharos* (*lighthouse*) Linux supercomputer at Stevens Institute of Technology, producing 4 forecast cycles of ensemble river discharge per day, simulated at hourly time step, that feed into the New York Harbor Observing and Prediction System (NYHOPS) (Bruno et al., 2006; Georgas et al., 2007; Georgas et al., 2014). NYHOPS was developed at Stevens Institute of Technology's Davidson Laboratory to generate forecasts of the Atlantic Coast, New York Harbor, and Hudson River region through in-situ monitoring equipment and hydrodynamic modeling (Blumberg et al., 2015).

### 2.2.1 HEC-HMS model description

The Hudson River Basin was modelled using the latest Hydrologic Engineering Center's Hydrologic Modeling System (HEC-HMS), version 4.1 (USACE, 2015). HEC-HMS, developed by the US Army Corps of Engineers, is a conceptual semi-distributed hydrological model that has been used extensively in rainfall-runoff modeling and other related hydrological studies (Anderson et al., 2002; Neary et al., 2004; Knebl et al., 2005; Amengual et al., 2009; Chu and Steinman, 2009; Halwatura and Najim, 2013; Meenu et al., 2013; Seyoum et al., 2013; Zhang et al., 2013; Yang and Yang, 2014). The model uses a number of adjustable empirically derived parameters that describe the overall structure of the basin including parameters for runoff, baseflow, and river routing (Feldman, 2000). In this work, the Modified Clark (ModClark) distributed method (Kull and Feldman, 1998) was used to account for the spatial variability and characteristics of the basin. The gridded precipitation inputs were used to enable spatially distributed infiltration calculations at all regions of the basin. Infiltration capacity in the model was quantified using the gridded curve number (CN) methodology derived by the Soil Conservation Service (SCS) (USDA, 1986; Mishra and Singh, 2013). The SCS CN method estimates precipitation excess as a function of cumulative precipitation, soil cover, land use and antecedent moisture content, total length of the river, and the elevation of the catchment area (Scharffenberg, 2015). The baseflow component of the model includes the initial flow and the recession constant to account for ground-water contributions to stream flow (Chow, 1959; Maidment, 1992; Feldman, 2000).

### 2.2.2 Hudson River Basin hydrological model datasets

The ArcGIS HEC-GeoHMS 10.2 extension (Fleming and Doan, 2013) was used to prepare and import the geographical information system (GIS) data into the HEC-HMS (Johnson et al., 2001). The regional model datasets shown in Figure 2 include topography obtained from the USGS National Elevation Dataset (NED) (Gesch et al., 2002), land surface cover obtained from the US Department of Agriculture National Resource Conservation Service (NRCS) and soil data for New York State and New Jersey gathered from the State Soil Geographic Database (STATSGO) (Miller and White, 1998). Land use datasets were obtained from the USGS National Land Cover Dataset (NLCD) (Homer et al., 2012).





The Hudson River basin was first delineated into sub-basins based on flow direction and accumulation derived from a digital elevation model (DEM) using HEC-GeoHMS (Fleming and Doan, 2013), and then each sub-basin was subdivided into hydrologic response units, each of which has a gridded curve number representing its runoff response rate based on its unique combination of land use, soil, and slope (Gassman et al., 2007). The gridded SCS curve number was obtained by intersecting land use and land cover with the soil data using the -CN-grid tool in HEC-GeoHMS. An example of the curve number grid GIS layer over a sub-basin of the Hudson River is shown in Figure 2. The other hydrologic parameters namely, imperviousness storage coefficient and imperviousness, were derived from the GIS datasets listed earlier this section (Figure 2). For observed river discharge we used fifteen USGS gauging stations that were made available through the National Water Information System (NWIS). The dataRetrieval R package (Hirsch and De Cicco, 2015) was used to download and process the USGS data into the R environment (R Core Team, 2012). We used river discharge data recorded at 15 minutes time intervals to form a complete HEC-DSSVue (HEC, 2009) database of streamflow observations over the study region (Figure 2). In total, we used 25-years of available historical flow data (not shown) to derive the model baseflow recession constants that are important to better simulate the falling limb of the discharge hydrograph for each sub-basin. More precisely, an automated base flow separation technique based on the R low flow statistics package "lfstat" was used (Gustard and Demuth, 2009; Koffler and Laaha, 2012). The optimal recession constant values for the sub-basins ranged from 0.67 to 0.90 depending on the sub-basin that was considered. These calculated values were consistent and in agreement with the ones reported in the literature (Pilgrim and Cordery, 1993; Feldman, 2000).

For the initial baseflow, we used observed conditions in the gauging stations to reduce uncertainties in the model as the model was intended to forecast short term extreme events within a 96-hr forecast horizon and not long term simulations. The model was forced with gridded precipitation, discussed in detail in the next sections of this paper, to work with the ModClark transform method (Kull and Feldman, 1998).

### 2.3 Model meteorological datasets

### 2.3.1 North American Regional Reanalysis (NARR)

NARR is a long term, dynamically consistent, high resolution, high frequency, atmospheric and land surface hydrology data set for the North American domain (Mesinger et al., 2006). NARR was developed as a major improvement upon the earlier National Centers for Environmental Prediction - National Centre for Atmospheric Research Global Reanalysis 1 (NCEP-NCAR GR1). NARR data has successfully assimilated high-quality and detailed precipitation observations into the atmospheric analysis to create a long-term, consistent, high-resolution climate dataset for the North American domain. The temporal resolution of the NARR data is 3 hours and the spatial resolution is 32 km (Mesinger et al., 2006). The NARR precipitation data has been used in a number of hydrological studies. For instance, Choi et al. (2009) used the NARR data sets to successfully calibrate the Semi-Distributed Land Use-based Runoff Process (SLURP) model. Solaiman and Simonovic



(2010) used the NARR data in a regional hydrological basin and reported satisfactory performance of such data in scarce regions.

In this work, NARR precipitation data, from the 26th to 31st of August 2011, corresponding to Hurricane Irene was used in the hydrological model applied to the Hudson River Basin in a 2-km common hydrologic gridded format using bicubic interpolation. Table 1 displays the rainfall accumulation totals extracted from the NARR data for selected sub-basins of the Hudson River. The hydrological simulation using this dataset of precipitation was considered as the simulation of reference and was compared with the ensemble forecast that is reported in the next section.

### 2.3.2 Global Ensemble Forecast System Reforecast (GEFS/R)

The Global Ensemble Forecast System (GEFS) is a weather forecast model made up of 21 ensemble members (Hamill et al., 2013; Hamill et al., 2015; Zhou and Zhu, 2016). The GEFS accounts for the amount of uncertainty in a forecast by generating an ensemble of multiple forecasts, each minutely different, or perturbed, from the control forecast. The GEFS, 1 degree horizontal resolution, data reforecasts used initial conditions obtained from high quality reanalyses data and same assimilation system that is used operationally. These reforecasts have been shown to be particularly useful for the calibration of relatively uncommon phenomena such as heavy precipitation (Hagedorn, 2008; Hamill et al., 2008). In relation to hydrology, reforecasts help produce quantitative probabilistic estimates of river streamflow that are as sharp and reliable as possible (Schaake et al., 2007).

### 2.4 Statistical criteria used to assess models performance

The performance of the models was statistically evaluated using the criteria of Nash-Sutcliffe Efficiency [referred hereafter as NSE (Eq. (1)] and Bias (in %) between simulations and observations [referred hereafter as BIAS Eq. (2)]. The NSE measures the fraction of the variance of the observed flows explained by the model in terms of the relative magnitude of the residual variance ('noise') to the variance of the flows ('information'); the optimal value is 1.0 and values should be larger than 0.0 to indicate 'minimally acceptable' performance (Nash and Sutcliffe, 1970; O'Connell et al., 1970).

$$NSE = 1 - \frac{\sum_{i=1}^{N}(P_i - O_i)^2}{\sum_{i=1}^{N}(O_i - \bar{O}_i)^2} \;-------- Eq(1)$$

where $N$ is the number of compared values, $P_i$ is the simulated (forecast) value, $O_i$ is the observed value and $\bar{O}_i$ is the average of $O_i$ time series.

The BIAS measures the average tendency of the simulated values to be larger or smaller than the observed ones. The optimal value of BIAS is 0.0, with low-magnitude values indicating accurate model simulation. Positive values indicate overestimation, whereas negative values indicate underestimation (Yapo et al., 1996).



$$BIAS(\%) = 100 \frac{\sum_{i=1}^{N}(P_i - O_i)}{\sum_{i=1}^{N} O_i} - - - - - - - - - - Eq(2)$$

Model BIAS is reported relative to the mean observation magnitude (Eq 2), in percentage (%).

## 3 Results

### 3.1 HEC-HMS model calibration using NARR precipitation data

Upon implementing the model set-up, we calibrated the hydrological model to assess its ability to reproduce the Hurricane Irene event using the North American Regional Reanalysis (NARR) gridded precipitation data (Mesinger et al., 2006). The HEC-HMS model was run on a 2x2 km Standard Hydrologic grid resolution (SHG) (Maidment and Djokic, 2000) at hourly time steps. The simulated flow hydrographs were calibrated against hourly river flow observations to obtain optimal performance in terms of both runoff volume and peak flow. The hydrological parameters were modified to produce a best-fit model using a root-mean-square error (RMSE) objective function within the HEC-HMS model's Nelder Mead optimization method (Barati, 2011; Seyoum et al., 2013), aiming at maximizing the fit between simulated streamflow and observations at fifteen U.S. Geological Survey (USGS) gauging stations (Figure 2). The calibration was also carried out by comparing visually and statistically to produce an accurate simulation of discharge at the Hudson River gauging stations (Figure 2). We also used the HEC-HMS uncertainty function which, through a variant of Latin Hypercube Sampling, varies sensitive model parameters within a defined range, thereby producing an estimate of best-fit parameters after multiple iterations (Mousavi et al., 2012). For example, the hydrologic Muskingum routing parameters were modified to include a greater ratio of attenuation to translation of runoff in the sub-basins which significantly improved the model results (Figure 5). To assess and evaluate the parameter uncertainties we performed a Monte-Carlo-based uncertainty analysis available in HEC-HMS 4.1 (Scharffenberg et al., 2015).

Table 1 lists the NARR accumulated precipitation for each sub-basin of the Hudson River while the summary of the model fit represented by the NSE and BIAS criteria is shown in Figure 4. The NARR hydrological model results (also referred to as the simulation of reference) showed a reasonable fit between model and observations in all the selected sub-basins. The lowest NSE was 0.75 and certain sub-basins had NSE values higher than 0.90 (Figure 4). Amongst the 15 flow stations examined in the study, 13 stations had a BIAS below 10% while 2 stations had a BIAS higher than 10%. It was observed that the stations with a higher BIAS were located in the upstream parts of the basin, notably the Hoosic River sub-basin (USGS ID 01334500) (Figure 4). Importantly, the hourly hydrograph shape and timing of peaks accurately replicated the observations as illustrated in Figure 5. Overall, the reference simulation exhibited a representative fit to observations. Thus, the developed calibrated framework showed promising results for generating 96-hour-lead streamflow forecasts using ensemble member weather forecast forcing.



## 3.2 Ensemble river discharge retrospective forecast

We forced the HEC-HMS model in ensemble mode, with precipitation fields from the GEFS-retrospective forecast ensemble members to examine the variations in simulated discharge among the ensemble members. More specifically, we fed the Hudson River basin hydrological model with every single GEFS member of the 21 available members. The resulting sets of streamflow

forecasts were then analysed to better understand the uncertainty of the streamflow forecasts that arises from the hydrologic framework's response to uncertainties in the meteorological forcing. The spread of ensemble members is considered as a useful measure of forecast uncertainty (Pappenberger et al., 2005).

To assess the skill of the forecasts, we compared, at lead times of 72, 48 and 24 hours, observations with results of the individual ensemble members, the median of all members, the ensemble control member (GEFSC00) and the NARR simulation of

reference (Figure 5). We chose to include the control member in these comparisons as a proxy for the single deterministic models used when ensemble forecasts are not considered. The hydrological parameters and baseflow conditions calibrated in the NARR simulation of reference were retained in all simulations. The 98$^{th}$ and 2$^{nd}$ percentiles of accumulated precipitation from the 21 ensemble members at each forecast are reported in Table 1. One notes the high uncertainty in the precipitation data from the weather ensemble 72-hours prior to the event which was translated by the hydrological model to a high

uncertainty in the simulated streamflow. For instance, in station 01390500 (Saddle River at Lodi, NJ), there was a 20-fold spread in the accumulated precipitation amongst the 21 ensemble members. For the same station the peak flow ranged from base flow (3 m$^3$/s) to 242 m$^3$/s (exceeding the major flood threshold) (Figure 5). At the Hackensack River at New Milford the peak flow was ranging anywhere between 20 m$^3$/s and 525 m$^3$/s. The magnitude of spread in other stations was similar (Figure 5). At that point of the retrospective forecast, the streamflow simulated using the control member (GEFSC00) underestimated

the observed flow in all stations except for station ID 01375000 (Croton River on Hudson, NY) where the flow was simulated correctly (Figure 5). However, the spread between the individual ensemble members remained very high as reported above. In terms of estimated time of peak, one notes that the GEFSC00 control member correctly projected the peak time of the event in all stations with an error of ±3 hours when compared with observations. However, other individual members had an offset of up to 24 hours between simulated and observed peaks in certain stations (5-a1, 5-b1, 5-c1, 5-d1 and 5-e1). For instance, at

station 01375000 (Croton River on Hudson, NY), one individual member was projecting an estimated peak on the evening of August 28 while another individual member was projecting it at noon of the following day. In that station the observed peak was around midnight of August 28. This finding offers an interesting perspective in terms of precisely extracting hydrograph features as one may, therefore use the control member at this stage of the forecasts to precisely project the time of the peak with a temporal error of ±3 hours. In terms of statistical evaluation, Figure 6 and Figure 7 report the NSE and BIAS (%) for

selected stations. For station 01375000, the control member (GEFSC00) predicts the flow accurately with a NSE of approximately 0.95, however the flow is underestimated by about 20% 72 hours prior to the event. Overall, only seven ensemble members at this station had a NSE above 0.70 while more than 60% of the members underestimated or overestimated the flow hydrograph by more than 30% (Figure 7). Although uncertainties in baseflow initial conditions and model



hydrological parameters are not addressed in this work, one may argue that uncertainties from precipitation inputs have a substantial impact on the prediction compared to remaining uncertainties in the initial conditions and parameters of the calibrated hydrological model.

In the next reforecast, 48 hours before the event, the spread (or the uncertainty envelope) was reduced substantially, notably in the projected time of the peak (Figure 5-a2, 5-b2, 5-c2, 5-d2 and 5-e2). This is primarily due to the decrease in accumulated precipitation ensemble spread (Table 1). The magnitude of the peak ensemble spread was between 104 to 229 $m^3/s$ for the Saddle River. The same river had a spread on the order of 20-fold in the prior forecast. This improvement was observed at other stations as well. At 48hrs lead time, the control member systematically overestimating the flow at all stations (e,g. Figure 5-a2, 5-b2, 5-c2, 5-d2 and 5-e2), but it consistently predicted the time of peak, as was the case in the forecast issued 72 hours before the event. The uncertainty in the time of the peak also decreased, with most of the ensemble members predicting the peaks around ±3 hours from the observed ones. This suggests that the peak ensemble spread may not be the only metric that should be analyzed to quantify potential uncertainty, it suggests that other features such as the peak timing and magnitude should also be examined. The control member (GEFSC00), which predicted the flow correctly for Croton River on Hudson (ID 01375000) 72 hours before the event, however, at 48hrs lead time showed a marked overestimation of the flow 65%. This suggests that the accuracy of the predictions also varies temporally and no single member can be relied upon consistently as a perfect forecast. Overall one notes that for 48 hours before the event there is a significant improvement in NSE for most of the members in the examined stations.

In the final reforecast considered, 24 hours before the event, the spread was further reduced with all the members predicting the rising and the falling limbs of the hydrograph more accurately. The control member presented an almost perfect forecast for Hackensack River at New Milford (Figure 6). The peak ensemble spread was reduced by 57%, 60% and 48% compared to the 72hrs-lead-time predicted peak ensemble spread at the Hackensack River, Saddle River and Wallkill River stations, respectively. Also, the peaks were predicted to occur within ±3 hours of the observed event peak in all cases. The control member seemed to be consistent with the observations in terms of the peak time at this stage of the forecast and the uncertainties are less than that projected 48 hours before the event. At this stage of the forecast 75% of the ensemble members had a NSE higher than 0.75 (Figure 6). This is consistent with findings in recent ensemble streamflow studies that used different modelling frameworks (Thielen et al., 2009; Fan et al., 2014; Yang and Yang, 2014). The results also suggest that the magnitude of spread between the ensemble members depends significantly on the sub-basin drainage area. For example, at station ID 01391500, Saddle River at Lodi (140 $km^2$) a peak discharge of 242 $m^3/s$ corresponds to a maximum accumulated precipitation of 206 mm. Thus, there is peak flow of about 1 $m^3/s$ for 1 mm of accumulated precipitation. This ratio however increases for a basin with a larger area. For example, at Wallkill River at Gardiner, NY (ID 01371500, 1800 $km^2$) there was a peak flow of approximately 10 $m^3/s$ for 1 mm of accumulated precipitation as it propagates (non-linearly) through the drainage area, thus there is a correlation between the spread in the precipitation data and the area of the sub-basin.

To have an overall assessment of the forecast skill, we calculated the range and average NSE and Bias (%) across all stations (Figure 8) in the Hudson River basin for the different reforecast ensemble members 24 hours before the event. The figures





depicts how a member that has a good NSE and BIAS at one station can have a very poor performance in other parts of the basin that may partly be due to the statistical downscaling of precipitation from 1 degree resolution to 2km, and the associated uncertainty in the spatial distribution of precipitation. The median of all the members and the control member showed a good performance with an average NSE of 0.75 compared to each of the members. The results show that there is no "one size fits all" solution for selecting an ensemble member, noting that each sub-basin has its own distinct set of characteristics manifested in local conditions such as the size of the basin and land use. This finding calls for further work involving higher resolution precipitation models to assess the effect of basin size and meteorological forcing resolutions.

### 3.2.1 Threshold exceedance persistence diagram

In addition to the statistical metrics and visual comparison of stream-flow, an assessment of the forecasts skill was carried out for Hurricane Irene using a threshold exceedance persistence diagram at 6-hr time intervals of the forecasts streamflow time series (Figure 9). The exceedance diagram used in this study is an adaptation of the operational European Flood Alert System (Bartholmes et al., 2009; Thielen et al., 2009). Such quantitative diagrams give an idea of the forecast persistence and support operational flood management decisions and human judgment. The threshold for each station was based on the "major flooding" flow defined by the NOAA National Weather Service as a category of flood where extensive inundation of structures and roads is expected, leading to significant evacuations of people and/or transfer of property to higher elevations is called (NWS, 2012) . The major thresholds for observed and simulated discharge were transformed into dichotomous time series of 1 (=Yes, the major threshold is exceeded) and 0 (=No, the major threshold is not exceeded). The information of each cell in the matrix diagram is the probability of exceeding the major flooding value, calculated using all the ensemble members at a given forecast. For instance, a probability of 100% suggests that all 21 ensemble members used in the work are projecting a major flood. Figure 9 exhibits the exceedance diagram results at selected stations of the Hudson River basin in which observations exceeded the major flood threshold during this event. The results suggest that one may establish a highly reliable streamflow forecast 48 hours prior to the event. For instance, the 72-hrs-lead-time reforecasts (26-Aug-2011 00:00 GMT) for station 01381900 were projecting a 52% probability (11 members out of 21) of having a major flood event 78 hours out. However, the probability increased to 100% in the 27-Aug-2011 00:00 GMT (48 hours before the event) reforecast for the same station. The average across all stations for the day-3 (72 hours before the event) reforecast was showing a 60% probability of having a major flood on 29-Aug-2011 (between 00:00 and 06:00 am GMT), while the day-2 (48 hours before the event) reforecasts had a 99% average chance of exceeding the major flood threshold,. This highlights the increase in reliability the event approaches. The diagram also suggests that the persistence of the event occurrence among subsequent forecasts is a good indicator to trigger a major flood warning, especially when the proportion of members above the threshold exceeds 71% (15 or more out of the 21 ensemble members). By contrast, station 0137500, both observations and flow ensemble members did not exceed the major flood threshold (Figure 9). This is particularly important for the reliability and validation of the operational forecast system. However, caution should be practiced when interpreting the persistence exceeding diagrams as time series will have to be examined in parallel to confirm any potential discrepancy between the models.



## 4. Summary and Discussion

The first part of this work consisted of implementing a regional scale hydrological modeling framework on of the Hudson River basin using the HEC-HMS model (USACE, 2015) forced with NARR gridded precipitation inputs (Mesinger et al., 2006). The second part investigated the use of GEFS (Hamill et al., 2013; Hamill et al., 2015) ensemble inputs to
retrospectively forecast an extreme hydrological event, Hurricane Irene, with a 96-hr time horizon at hourly time step. In total, 21 GEFS ensemble members were tested on the Hudson River basin at 72, 48, and 24 hours prior to the event.

This work did not address uncertainty in hydrological initial conditions and model parameters, which were optimized in the NARR-based calibration, but rather was focused on how dominant the precipitation inputs are on the results of the hydrological streamflow forecasts.

In terms of assessment, we visually and statistically quantified the results of the GEFS individual members, the deterministic forecasts represented by the control member (GEFSC00), and the median of all members. We also used the persistence exceedance diagram of established thresholds to project the possibility of major floods based on all the individual members. The visual comparison and the statistical metrics show that ensemble forecasts are advantageous in quantifying uncertainty of forecasts lead time and raising reliability from an operational perspective. The work shows that streamflow forecasts are highly
dependent on the meteorological inputs and reflect uncertainties associated with these inputs. Clearly, the relatively small area of the sub-basin and resolution of the weather model was of importance for the spread overestimation due to the precipitation inputs. In this context, higher resolution weather models such as the European Centre for Medium‑Range Weather Forecast (ECMWF) (Molteni et al., 1996; Thiemig et al., 2015) or the Coupled Ocean-Atmosphere Mesoscale Prediction System (COAMPS)® (Pullen et al., 2015) may be of particular interest in smaller scale sub-catchments in order to provide higher
accuracy. The operational GEFS ensemble forecast datasets were upgraded to 0.5 degrees resolution on December 2, 2015 (http://www.nco.ncep.noaa.gov/pmb/changes/). More detailed weather model forcing should lead to a reduction in the streamflow spread, and reduce the uncertainty of the forecast.

The presented hydrologic modeling of the Hudson River basin demonstrates that a given streamflow ensemble member may produce streamflow that is highly in agreement with observations, accounting for uncertainty in precipitation and in
hydrologic-model parameters. However, there is no "one size fits all" solution for selecting an ensemble member, noting that each sub-basin has its own distinct set of characteristics manifested in local conditions such as the area of the drainage basin and land use. Furthermore, caution should be exercised in forecast models that only use the control member from the weather models or the average ensemble member as it may lead to considerable deviation in river discharge forecasts from observations resulting in false warnings and missed flooding events, thereby, decreasing the potential reduction of flood risk (Figure 5).
The findings also suggest that higher confidence in the river discharge forecasts may be attained as we approach a major event by approximately 48 hours. The outcomes of this work provide interesting perspectives for future ensemble post-processing techniques and features extraction, notably regarding the peak timing of an extreme hydrologic event when combined with the major flood persistence diagram.





This operational framework offers an improvement over the available NOAA's Advanced Hydrological Prediction System (AHPS) in this particular region (McEnery et al., 2005). The AHPS streamflow forecasts are at 6-hour time interval using one weather control member for input and with a lead time that is less than 60 hours in this region (Adams, 2015).

The Hudson River basin regional scale model may be potentially used for numerous applications such as continuously

forecasting the overall variability of the water resources (Saleh et al., 2011; Pryet et al., 2015; Saraiva Okello et al., 2015), predicting fate and transport of water quality such as nitrate (Schoonover and Lockaby, 2006; Wang et al., 2012; Bastola and Misra, 2015; Schuetz et al., 2015) and climate change scenarios (Ducharne et al., 2007; Graham et al., 2007; Ducharne et al., 2010; Quintana Seguí et al., 2010; Habets et al., 2013). Moreover, socio-economic analysis may be used to weigh how such improved forecasts potentially prevent loss of life and minimize the damage to property, with the aid of effective

communication and social media.

**Acknowledgments**

This work was funded by a research task agreement entered between the Trustees of the Stevens Institute of Technology and the Port Authority of New York and New Jersey, effective August 19, 2014. We are grateful to Dr. William Scharffenberg and his team at the USACE Hydrologic Engineering Center for their help in operationally running HEC-HMS and HEC-DSSvue

on Linux. We are grateful to Ms. Sarah R. Hatala from New Jersey Department of Environmental Protection Office of Engineering and Construction Bureau of Dam Safety and Flood Control for providing reservoir data. We are thankful to Mr. Alon Dominitz from New York State Department of Environmental Conservation for providing us with sources of dams elevation-storage data. We are also thankful to Mr. Jacob Isleib from the USDA-Natural Resources Conservation Service for his help in extracting the soil data for New York and New Jersey from the State Soil Geographic Database (STATSGO). We

are thankful to Mr. Wesley Ebisuzaki from the NOAA Climate Prediction Center for his help with the NARR data. The GEFS Retrospective Forecasts were graciously provided by Drs. Yuejian Zhu, Justin Cooke, Xiaqiong Zhou, and Rebecca Cosgrove of NOAA EMC.



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



# A Retrospective Streamflow Ensemble Forecast for an Extreme Hydrologic Event: a Case Study of Hurricane Irene and on the Hudson River basin

5                                              **-Tables-**

| Table 1 Percentiles of total accumulated precipitation from the 21 GEFS ensemble members along with the peak discharge for forecasts issued at lead times of 72 to 24 hours. The drainage area of each basin is also reported ||||||||
|---|---|---|---|---|---|---|---|
| **Station USGS ID / name** | **Basin area (km²)** | **Forecast date** | **NARR precipitation (mm) over the whole event** | **GEFS precipitation (mm) over the whole event** | | **Simulated peak flow (m³/s)** | |
| | | | | **2ⁿᵈ** | **98ᵗʰ** | **2ⁿᵈ** | **98ᵗʰ** |
| **01391500 Saddle River at Lodi, NJ** | 141 | 26-Aug-11 | 143 | 11 | 206 | 3 | 242 |
| | | 27-Aug-11 | | 110 | 201 | 104 | 229 |
| | | 28-Aug-11 | | 108 | 178 | 105 | 200 |
| **01378500 Hackensack River at New Milford, NJ** | 293 | 26-Aug-11 | 143 | 11 | 206 | 20 | 525 |
| | | 27-Aug-11 | | 108 | 200 | 196 | 514 |
| | | 28-Aug-11 | | 111 | 183 | 225 | 442 |
| **01371500 Wallkill River at Gardiner, NY** | 1800 | 26-Aug-11 | 106 | 3 | 190 | 22 | 1990 |
| | | 27-Aug-11 | | 80 | 187 | 475 | 1857 |
| | | 28-Aug-11 | | 82 | 154 | 558 | 1585 |
| **01388500 Pompton River at Pompton Plains, NJ** | 329 | 26-Aug-11 | 130 | 7 | 202 | 156 | 1208 |
| | | 27-Aug-11 | | 101 | 200 | 474 | 1150 |
| | | 28-Aug-11 | | 98 | 168 | 490 | 1024 |
| **01375000 Croton River on Hudson, NY** | 979 | 26-Aug-11 | 126 | 13 | 192 | 2 | 1255 |
| | | 27-Aug-11 | | 123 | 190 | 604 | 1205 |
| | | 28-Aug-11 | | 112 | 182 | 503 | 1205 |



# A Retrospective Streamflow Ensemble Forecast for an Extreme Hydrologic Event: a Case Study of Hurricane Irene and on the Hudson River basin

5                                               -Figures-



Figure 1. Geostationary Operational Environmental Satellite (GOES) East image of Hurricane Irene making landfall on August 28, 2011 (image source: National Oceanic and Atmospheric Administration, 2011).





Figure 2 Map showing the Hudson River Basin topography including basin divisions (thin black lines) and hydrographic network. Examples of land use, curve number, and imperviousness datasets (zoomed) that were used in HEC-GeoHMS to construct the hydrological model are also shown. The upper right side of the figure exhibits an example of the HEC-HMS model structure and its sub-basins using Standard Hydrologic Grids [SHG (2*2 km)].





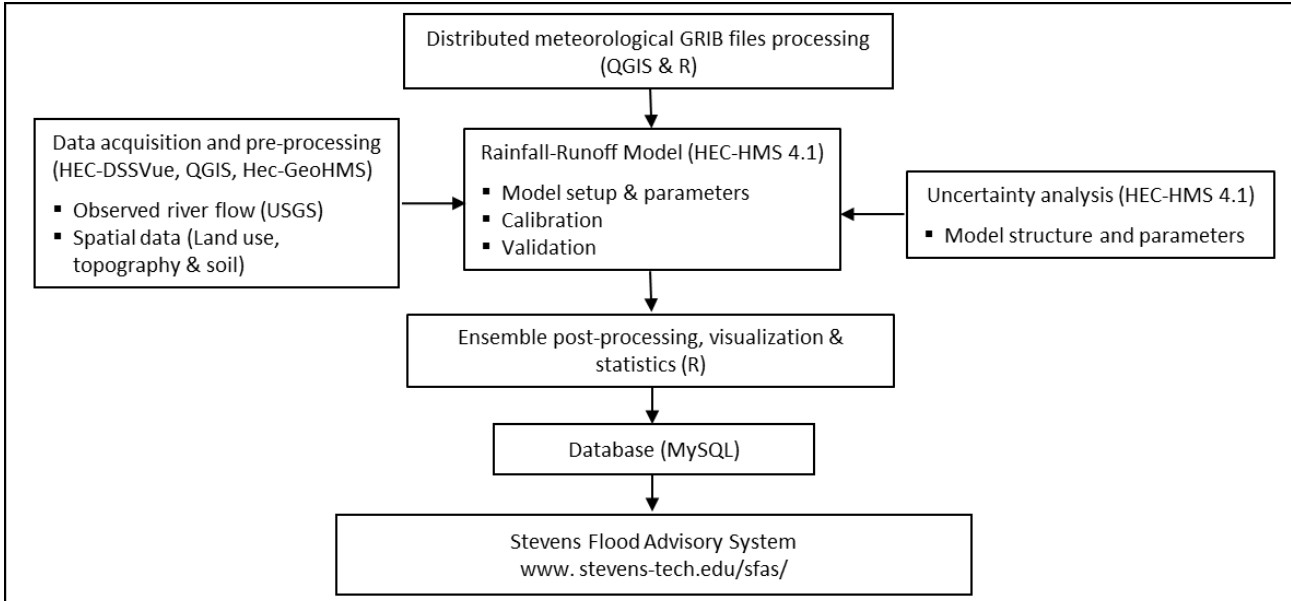

Figure 3 Framework design steps including spatial data and USGS discharge time series. Processing of General Regularly-distributed Information in Binary GRIB precipitation files data was carries out in Qgis (QGis, 2011) R and Python and exported to the HEC-DSSvue storage system. Basin parameters were derived for the study area and the hydrological model was run using these inputs.





Figure 4 Summary of HEC-HMS performances at the USGS streamflow stations. The circles represent the criteria of Nash and Sutcliffe (NSE) while the squares represent the Bias (%). The statistical criteria are computed at an hourly time step. The model performances are illustrated for the NARR precipitation forcing. The map also shows the total observed accumulated precipitation received in 48 hours during August 28–29, 2011 (data from National Oceanic and Atmospheric Administration, 2011) .The time series for selected flow station are presented in Figure 5.





Figure 5 Models performance in selected station at lead times of 72 h, 48 h and 24 hours from the observed peak flow, reported time is in UTC.





Figure 6 Models performance represented by Nash-Sutcliffe efficiency (NSE) statistical metric at lead times of 72 h, 48 h and 24 hours from the observed peak flow. The metrics are also showing the NARR model outputs and the median of ensemble members of this station.





Figure 7 Models performance represented by Bias (%) statistical metric at lead times of 72 h, 48 h and 24 hours from the observed peak flow. The metrics are also showing the NARR model outputs and the median of ensemble members of this station.





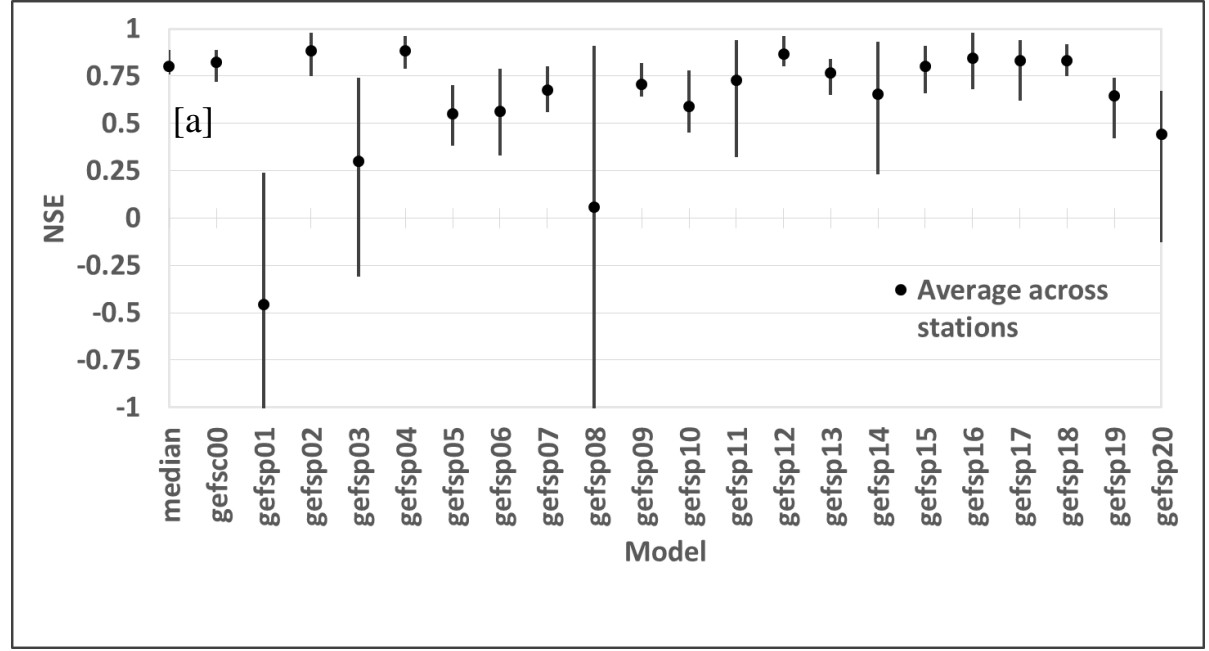

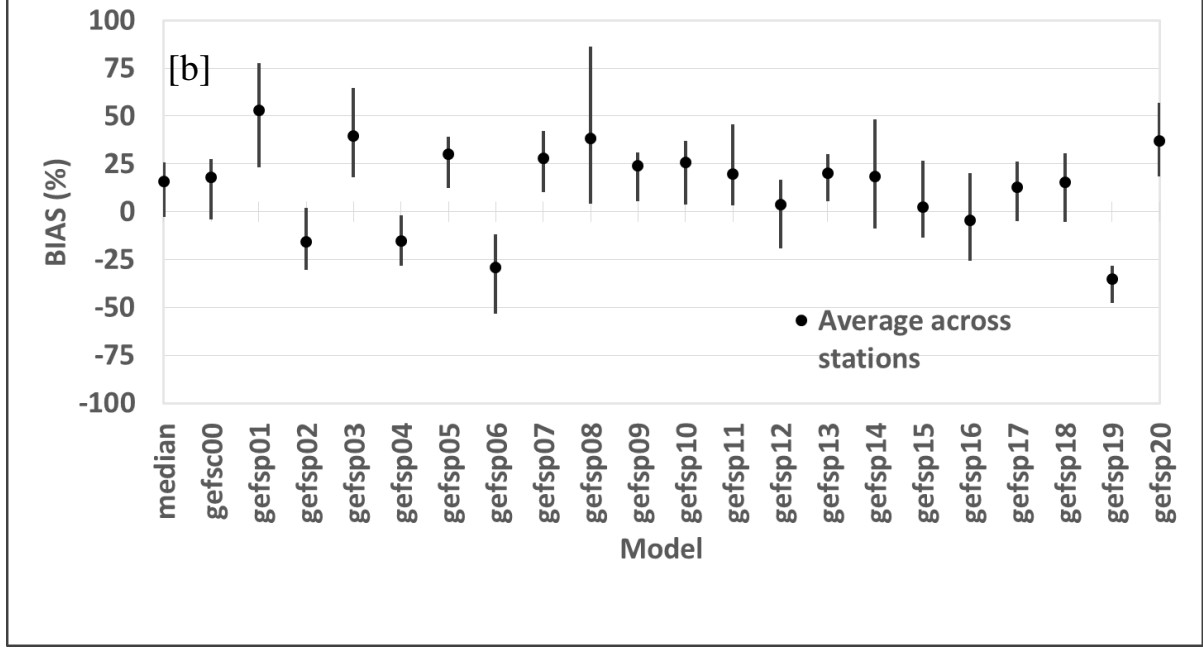

5    Figure 8 Overall NSE and BIAS (%) across the stations for forecast issued 24 hours before Hurricane Irene.



Hydrology and Earth System Sciences — Open Access — Discussions — EGU

| Station & major flood threshold | Forecast hr | 8/26/2011 | | | | 8/27/2011 | | | | 8/28/2011 | | | | 8/29/2011 | | | | 8/30/2011 | | | | 8/31/2011 | | | |
| --- | --- | --- | --- | --- | --- | --- | --- | --- | --- | --- | --- | --- | --- | --- | --- | --- | --- | --- | --- | --- | --- | --- | --- | --- | --- |
| | | 0:00 | 6:00 | 12:00 | 18:00 | 0:00 | 6:00 | 12:00 | 18:00 | 0:00 | 6:00 | 12:00 | 18:00 | 0:00 | 6:00 | 12:00 | 18:00 | 0:00 | 6:00 | 12:00 | 18:00 | 0:00 | 6:00 | 12:00 | 18:00 |
| Hoosic River at Eagle Bridge, NY (01334500) 886 m³/s | 08/26/2011 t00 | 0 | 0 | 0 | 0 | 0 | 0 | 0 | 0 | 0 | 0 | 0 | 14 | 52 | 71 | 62 | 29 | | | | | | | | |
| | 08/27/2011 t00 | | | | | 0 | 0 | 0 | 0 | 0 | 0 | 0 | 62 | 100 | 100 | 86 | 29 | 0 | 0 | 0 | 0 | | | | |
| | 08/28/2011 t00 | | | | | | | | | 0 | 0 | 0 | 71 | 100 | 100 | 90 | 0 | 0 | 0 | 0 | 0 | 0 | 0 | 0 | 0 |
| | Observed | | | | | | | | | | | | | | | | | | | | | | | | |
| Wappinger Creek at Wappingers Falls, NY (01372500) 286 m3/s | 08/26/2011 t00 | 0 | 0 | 0 | 0 | 0 | 0 | 0 | 0 | 0 | 0 | 0 | 5 | 29 | 52 | 52 | 38 | | | | | | | | |
| | 08/27/2011 t00 | | | | | 0 | 0 | 0 | 0 | 0 | 0 | 0 | 10 | 86 | 90 | 86 | 48 | 5 | 0 | 0 | 0 | | | | |
| | 08/28/2011 t00 | | | | | | | | | 0 | 0 | 0 | 5 | 86 | 90 | 86 | 62 | 0 | 0 | 0 | 0 | 0 | 0 | 0 | 0 |
| | Observed | | | | | | | | | | | | | | | | | | | | | | | | |
| Hackensack River at New Milford, NJ (01378500) 158 m3/s | 08/26/2011 t00 | 0 | 0 | 0 | 0 | 0 | 0 | 0 | 0 | 0 | 0 | 0 | 14 | 52 | 67 | 62 | 48 | 33 | | | | | | | |
| | 08/27/2011 t00 | | | | | 0 | 0 | 0 | 0 | 0 | 0 | 0 | 62 | 100 | 100 | 95 | 86 | 48 | 5 | 0 | 0 | | | | |
| | 08/28/2011 t00 | | | | | | | | | 0 | 0 | 0 | 43 | 100 | 100 | 100 | 81 | 14 | 0 | 0 | 0 | 0 | 0 | 0 | 0 |
| | Observed | | | | | | | | | | | | | | | | | | | | | | | | |
| Passaic River at Little Falls, NJ (01389500) 283 m3/s | 08/26/2011 t00 | 0 | 0 | 0 | 0 | 0 | 0 | 0 | 0 | 0 | 0 | 0 | 5 | 24 | 67 | 67 | 67 | 67 | | | | | | | |
| | 08/27/2011 t00 | | | | | 0 | 0 | 0 | 0 | 0 | 0 | 0 | 19 | 90 | 100 | 100 | 100 | 100 | 100 | 100 | 100 | | | | |
| | 08/28/2011 t00 | | | | | | | | | 0 | 0 | 0 | 5 | 86 | 100 | 100 | 100 | 100 | 100 | 100 | 100 | 100 | 100 | 100 | 100 |
| | Observed | | | | | | | | | | | | | | | | | | | | | | | | |
| Rondout Creek at Rosendale, NJ (01367500) 708 m3/s | 08/26/2011 t00 | 0 | 0 | 0 | 0 | 0 | 0 | 0 | 0 | 0 | 0 | 0 | 19 | 48 | 57 | 48 | 38 | 19 | | | | | | | |
| | 08/27/2011 t00 | | | | | 0 | 0 | 0 | 0 | 0 | 0 | 0 | 76 | 95 | 100 | 81 | 29 | 5 | 0 | 0 | 0 | | | | |
| | 08/28/2011 t00 | | | | | | | | | 0 | 0 | 0 | 71 | 100 | 95 | 86 | 29 | 0 | 0 | 0 | 0 | 0 | 0 | 0 | 0 |
| | Observed | | | | | | | | | | | | | | | | | | | | | | | | |
| Saddle River at Lodi, NJ (01391500) 80 m3/s | 08/26/2011 t00 | 0 | 0 | 0 | 0 | 0 | 0 | 0 | 0 | 0 | 0 | 5 | 29 | 67 | 71 | 71 | 52 | 38 | | | | | | | |
| | 08/27/2011 t00 | | | | | 0 | 0 | 0 | 0 | 0 | 0 | 0 | 10 | 90 | 100 | 100 | 100 | 90 | 48 | 0 | 0 | | | | |
| | 08/28/2011 t00 | | | | | | | | | 0 | 0 | 0 | 81 | 100 | 100 | 100 | 100 | 81 | 5 | 0 | 0 | 0 | 0 | 0 | 0 |
| | Observed | | | | | | | | | | | | | | | | | | | | | | | | |
| Passaic River at Pine Brook, NJ (01381900) 141 m3/s | 08/26/2011 t00 | 0 | 0 | 0 | 0 | 0 | 0 | 0 | 0 | 0 | 0 | 0 | 0 | 19 | 52 | 52 | 57 | 57 | | | | | | | |
| | 08/27/2011 t00 | | | | | 0 | 0 | 0 | 0 | 0 | 0 | 0 | 19 | 81 | 100 | 100 | 100 | 100 | 100 | 100 | 100 | | | | |
| | 08/28/2011 t00 | | | | | | | | | 0 | 0 | 0 | 5 | 48 | 95 | 100 | 100 | 100 | 100 | 100 | 100 | 100 | 100 | 100 | 100 |
| | Observed | | | | | | | | | | | | | | | | | | | | | | | | |
| Pompton River at Pompton Plains, NJ (01388500) 329 m3/s | 08/26/2011 t00 | 0 | 0 | 0 | 0 | 0 | 0 | 0 | 0 | 0 | 0 | 0 | 14 | 43 | 71 | 67 | 67 | 62 | | | | | | | |
| | 08/27/2011 t00 | | | | | 0 | 0 | 0 | 0 | 0 | 0 | 0 | 57 | 95 | 100 | 100 | 100 | 90 | 76 | 57 | 19 | | | | |
| | 08/28/2011 t00 | | | | | | | | | 0 | 0 | 0 | 29 | 100 | 100 | 100 | 100 | 81 | 76 | 19 | 5 | 0 | 0 | 0 | 0 |
| | Observed | | | | | | | | | | | | | | | | | | | | | | | | |
| Raritan River at Bound Brook, NJ (01403060) 923 m3/s | 08/26/2011 t00 | 0 | 0 | 0 | 0 | 0 | 0 | 0 | 0 | 0 | 0 | 33 | 67 | 90 | 90 | 81 | 71 | 52 | | | | | | | |
| | 08/27/2011 t00 | | | | | 0 | 0 | 0 | 0 | 0 | 0 | 0 | 62 | 100 | 100 | 100 | 100 | 100 | 62 | 5 | 0 | | | | |
| | 08/28/2011 t00 | | | | | | | | | 0 | 0 | 100 | 100 | 100 | 100 | 100 | 100 | 81 | 5 | 0 | 0 | 0 | 0 | 0 | 0 |
| | Observed | | | | | | | | | | | | | | | | | | | | | | | | |
| Esopus Creek at Mount Marion, NY (01364500) 547 m3/s | 08/26/2011 t00 | 0 | 0 | 0 | 0 | 0 | 0 | 0 | 0 | 0 | 0 | 14 | 29 | 52 | 52 | 48 | 48 | 48 | | | | | | | |
| | 08/27/2011 t00 | | | | | 0 | 0 | 0 | 0 | 0 | 0 | 0 | 38 | 95 | 95 | 95 | 95 | 90 | 81 | 67 | 57 | 48 | | | |
| | 08/28/2011 t00 | | | | | | | | | 0 | 33 | 100 | 100 | 100 | 90 | 86 | 86 | 71 | 62 | 57 | 48 | 29 | 14 | 5 | 0 |
| | Observed | | | | | | | | | | | | | | | | | | | | | | | | |
| Mohawk River at Cohoes, NY (01375000) 1274 m3/s | 08/26/2011 t00 | 0 | 0 | 0 | 0 | 0 | 0 | 0 | 0 | 0 | 0 | 0 | 0 | 0 | 0 | 0 | 0 | | | | | | | | |
| | 08/27/2011 t00 | | | | | 0 | 0 | 0 | 0 | 0 | 0 | 0 | 0 | 0 | 0 | 0 | 0 | 0 | 0 | 0 | 0 | | | | |
| | 08/28/2011 t00 | | | | | | | | | 0 | 0 | 0 | 0 | 0 | 0 | 0 | 0 | 0 | 0 | 0 | 0 | 0 | 0 | 0 | 0 |
| | Observed | | | | | | | | | | | | | | | | | | | | | | | | |

Figure 9 Color-coded threshold exceedance diagram for Hurricane Irene forecasts at 6-hour intervals using the major flood threshold for each USGS station. The flow values below the station ID are the major flood threshold for a given station. The x-axis represents the 96-hr forecast horizon from the simulation date shown in the left column. The time period, in which the major event hydrological record exceeds the equivalent alert threshold, is indicated using a dark red cell while the cell values refer to the percentage of ensemble members that were projecting a major flood. For instance, if the value is 100 then all 21 flow ensemble members are projecting a major event within a given time interval, while 0 means none of the ensemble members are exceeding the major event threshold. The observed occurrence of the threshold is exhibited at the last row for each station, the red colour-code indicates an observed flow higher than the major flood threshold, while the green cell is flow below the major threshold. Reported time is in UTC.