# Peer review of "A Retrospective Streamflow Ensemble Forecast for an Extreme Hydrologic Event: a Case Study of Hurricane Irene and on the Hudson River basin"

_Hydrology and Earth System Sciences, 2016_

## Referee Comment (RC1) · Anonymous Referee #1 · 21 Apr 2016

GENERAL COMMENTS

This is very interesting work and a well-done study. I believe that this work constitutes a substantial contribution to scientific progress.

SPECIFIC COMMENTS

Comment 1: In the abstract the authors state that this modeling framework could be applied anywhere in the world. However, they use NARR dataset and a database from US based gage sites for calibration. How would these methods be applied for watersheds without gaging stations (or with only a few) outside of the US where the NARR dataset does not apply? Also, this statement was not discussed in the paper.

Comment 2: The HEC-HMS model uses the SCS Curve Number method that includes "antecedent moisture content" (P5 Line 22) as a parameter for estimating runoff. From my experience, model runoff estimation can be very sensitive to soil moisture. This indicates that calibrating the model will only produce accurate answers for the conditions of the storm it was calibrated to. How do you account for changing soil moisture in the forecast framework?

Comment 3: How long does it take for the streamflow forecasts to be produced? How much lead-time is left over? Is it enough to issue a warning?

TECHNICAL CORRECTIONS

P2 Line 18 – Extra period

P2 Line 31 – Change "using every ensemble" to "every member in the ensemble"

P3 Line 13 – "the ensemble spread was found . . ." – by who? Was that Komma et al. (2007)? I believe it would be better not to use passive voice in this instance.

P3 Lines 14-15 – Again with the passive voice – who devised it?

P5 Line 6 – change to "at hourly time steps"

P5 Lines 4-6 – Long sentence. Consider revising and dividing into two sentences.

P5 Line 22 – "soil cover, land use and antecedent moisture content" should be "soil type, antecedent soil moisture content, land use"

P6 Line 4 – should be "land use, soil type, and slope"

P6 Lines 6-7 – was it intentional to use "imperviousness" twice?

P10 Line 7 – Revise sentence – wording is unclear.

P10 Lines 8-10 – Check grammar and consider dividing sentence.

P10 Lines 11-13 – Consider rewording sentence and dividing.

P10 Line 16 – going into 3rd person – may want to avoid that

P10 Line 34 – Should "figures" be "figure"?

Figures 6&7 - They have too much information. Much of the information is lost due to its size and being squished into the other data. Consider simplifying or using alternate method to display information. Also, which member is the control? Cannot see it. Consider highlighting in some way.

P11 Line 3 – Change "showed a good" to "showed good"

P11 Lines 13-16 – Consider dividing sentence. Also, note extra space before period at end of sentence.

ADDITIONAL INFO

The authors may be interested in an upcoming publication on ensemble forecasting using ECMWF datasets listed here (http://rapid-hub.org/publications.html):

Snow, Alan D., Scott D. Christensen, Nathan R. Swain, James Nelson, Daniel P. Ames, Norman L. Jones, Deng Ding, Nawajish Noman, Cédric H. David, Florian Pappenberger (In Press), "A High-Resolution National-Scale Hydrologic Forecast System from a Global Ensemble Land Surface Model", Journal of the American Water Resources Association.

---

## Author Comment (AC1) · 22 Apr 2016

Dear Referee,

We are deeply grateful for your valuable comments and suggestions that will greatly improve the quality of this manuscript. We outline below your comments and our responses.

SPECIFIC COMMENTS Comment 1: In the abstract the authors state that this modeling framework could be applied anywhere in the world. However, they use NARR dataset and a database from US based gage sites for calibration. How would these methods be applied for watersheds without gaging stations (or with only a few) out-

[Figure]

side of the US where the NARR dataset does not apply? Also, this statement was not discussed in the paper.

Thank you for pointing this out. We will make sure that this statement is addressed properly in the paper. It is possible to use other sources of atmospheric data in the framework instead of NARR in order to apply the framework to watersheds in other countries. For instance, one may use atmospheric reanalysis products from the European Center for Medium range Weather Forecasting (ECMWF), National Centers for Environmental Prediction (NCEP) and National Center for Atmospheric Research (NCAR). The framework directly handles GRIB1, GRIB2 and NetCDF. As for watersheds without gaging stations (or with only a few), it is possible to use remote sensing river discharge data to calibrate and validate the modeling outputs. Despite that fact that such data have uncertainties, there have been many advancements in this field and there is potential for future applications (e.g., the Surface Water and Ocean Topography (SWOT) satellite mission).

Comment 2: The HEC-HMS model uses the SCS Curve Number method that includes "antecedent moisture content" (P5 Line 22) as a parameter for estimating runoff. From my experience, model runoff estimation can be very sensitive to soil moisture. This indicates that calibrating the model will only produce accurate answers for the conditions of the storm it was calibrated to. How do you account for changing soil moisture in the forecast framework?

This is a very important point that the referee is addressing. We are aware of the limitations in using static parameters for the SCS curve number method. To this end, the framework has a look up table for the initial abstraction parameters based on the hindcast and the continuous run of the model with the NARR data. We are actively working on integrating a machine learning technique which automatically selects the optimal initial abstraction parameters on the fly, this is a subject of active research by our team. We will make sure this is discussed in the paper.

Comment 3: How long does it take for the streamflow forecasts to be produced? How much lead-time is left over? Is it enough to issue a warning?

For the entire Hudson River Basin, the required time for GEFS (21 members) is around 30 minutes. This includes processing the GRIB files and post-processing of the ensemble outputs. We are currently running 125 ensemble members in the framework and this includes (in addition to GEFS) the ECMWF, ECMWF-HRES, the Short-Range Ensemble Forecast (SREF), the Canadian Meteorological Centre (CMC) and the North American Mesoscale Forecast System (NAM). The total time for all these ensemble members is around five and a half hours from pre-processing to updating the database and the website. We are currently updating the forecasts every 6 hours. The current lead time (87 hours) is sufficient for issuing a flood warning.

TECHNICAL CORRECTIONS

We are very grateful for the technical corrections, we will make sure they are all addressed in the manuscript.

ADDITIONAL INFO The authors may be interested in an upcoming publication on ensemble forecasting using ECMWF datasets listed here (http://rapid-hub.org/publications.html): Snow, Alan D., Scott D. Christensen, Nathan R. Swain, James Nelson, Daniel P. Ames, Norman L. Jones, Deng Ding, Nawajish Noman, Cédric H. David, Florian Pappenberger (In Press), "A High-Resolution National-Scale Hydrologic Forecast System from a Global Ensemble Land Surface Model", Journal of the American Water Resources Association.

Indeed, we are very interested in the Snow et al. (2016) publication once it becomes available online. Actually, I am very familiar with the work of certain co-authors and have also used the routing model RAPID in the past.

---

## Short Comment (SC1) · 13 May 2016

Subject: Scenario analysis for streamflow precasting

I've read with interest the executive–style summary of a comprehensive analysis of a recent extreme flood event. The storm and flood data from the retrospective ensemble streamflow forecast using HEC-HMS model for Hurricane Irene on the Hudson River Basin are summarized in Table 1 of the Discussion Paper.

I would call the "Retrospective" analysis also a "Scenario" one for streamflow

pre–(fore)casting, even though theirs is only for one storm on one basin using one hydrologic model.

It is virtually impossible to reproduce a past storm and flood event, an example of this being their 21 different precipitation reforecast datasets. The best that one can do is to capture its salient features and plan for the next bigger ones.

In contrast to the modified Clark unit hydrograph method in the semi–distributed HEC–HMS model, I suggest for consideration a lumped, though nonlinear, rainfall excess – direct runoff module. This is typified by a variable instantaneous unit hydrograph (vIUH) model of the 1974 vintage (Ding, 1974, 2011; Jun, 1989; Stanescu and Musy, 2006).

In hindsight, the concept of a nonlinear watershed response was first captured by Childs (1958) in a study on an earlier hurricane ("Diane") on nearby basins in New England. In it, he showed a family of observed nonlinear unit hydrographs for the Naugatuck River at Thomaston in Connecticut, which was reprinted in Ding (2011, Figure 2). His illuminating diagram was available both in a conference preprint and later a journal paper. From its very beginnings, however, this, to me, visionary work seemed to have fallen off our collective radar screen.

The hindcast data in their Table 1 enable an initial calibration, for the five sub-basins, of a 2011 variety of the vIUH model. This was a product of the Manning friction law, and had only one parameter. For calibration, in addition to the rainfall–excess data, this requires only the time to the flood peak and/or its magnitude, all observed, estimated, simulated, or a combination thereof.

[Figure]

This nonlinear rational–type formula for peak flow prediction (Ding, 2011, Eqs. 28 and 29) is as follows:

$$Q(j_p) = 0.2c(R_E/\Delta t)^{1.4}A\Delta t \ ,$$

$$j_p = 0.5 + \frac{0.535}{c(R_E/\Delta t)^{0.4}\Delta t} \ ,$$

in which:

$Q(j_p)$ = peak flow $(m^3/s)$,

$j_p$ = peak time $(\Delta t)$,

$A$ = basin area $(km^2)$,

$\Delta t$ = timestep size $(h)$; also the duration of the rainfall–excess storm,

$R_E$ = rainfall excess $(mm)$,

$c$ = scale parameter $((mm \ h^{-1})^{0.6})$.

For this Short Comment, only the short, 24-hour forecast lead–time is considered. The major assumptions made are:

- There was no distinction made between the rainfall and the rainfall–excess, i.e.
the basin was fully saturated and the infiltration losses negligible, and

- The rainfall–excess hyetograph was uniformly distributed in time.

New Tables 1 and 2 show, respectively, the input data for and output data from a step–by–step calculation for the vIUH parameter $c$ values and the model's peak times for five sub–basins. These peak times are longer than the "fixed" peak time of 24 h, i.e. $0.5\Delta t$, in the Discussion Paper.

The scale parameter $c$ value of 0.059 for the Prompton River is seen more than twice the rest. Figure 2 of the Discussion Paper indicates that this is a downstream-most basin having apparently a highest imperviousness or urbanization ratio. Through the lens of the vIUH model, the Prompton River was flahier than the rest.

The 1–parameter vIUH model assumes a nonlinear storage–discharge relation of the form: $Q = c^{1.67}S^{1.67}$, where $S$ is the water storage $(mm)$. This is in contrast to the linear relation, $S = RQ$, in the 2–parameter Clark synthetic unit hydrograph method, where $R$ is a storage coefficient $(h)$. (The Clark method has a second parameter $t_c$, the time of concentration, e.g. Straub et al., 2000).

The vIUH scale parameter $c$ and the Clark coefficient $R$ can be made related to each other by equating $Q$ in these two relations. This gives: $c = (1/R^{0.6})(1/S^{0.4})$. In a unit hydrograph, the storage $S$ is a variable represented by a recession curve starting from a maximun "unit amount" of the rainfall excess. Further comparative analysis using, for example, the statistical moments matching method (e.g. Ding, 1974, page 63) will be productive, such as determining the amount of storage remaining at the peak time especially by the Clark method. But this is beyond the scope of this Short Statement.

[Figure]

Lastly, the initial parameter $c$ value is re–calibratable (i.e. updatable) real–time from new observations using the Kalman filter, though I've had no personal experience implementing one.

References

Childs, E. F.: Northeastern floods of 1955: flood control hydrology. Journal of the Hydraulics Division, 84(3), pp.1-24, 1958.

Ding, J. Y.: Variable unit hydrograph. Journal of Hydrology, 22(1), pp.53-69, 1974.

Ding, J. Y.: A measure of watershed nonlinearity: interpreting a variable instantaneous unit hydrograph model on two vastly different sized watersheds. Hydrology and Earth System Sciences, 15(1), pp.405-423, 2011.

Jun, X.: Parameter identifiability of hydrological models with implicit structure: a numerical approach. Hydrological sciences journal, 34(1), pp.1-19, 1989.

Saleh, F., Ramaswamy, V., Georgas, N., Blumberg, A., and Pullen, J.: A Retrospective Streamflow Ensemble Forecast for an Extreme Hydrologic Event: a Case Study of Hurricane Irene and on the Hudson River basin, Hydrol. Earth Syst. Sci. Discuss., doi:10.5194/hess-2016-104, in review, 2016.

Stanescu, V. A. and Musy, A.: The non–linear unit hydrograph, in: VICAIRE, Module 1b: Engineering hydrology, Chapter 4: Transfer function, Section 4.4, 2006. http://echo2.epfl.ch/VICAIRE/mod_1b/chapt_4/main.htm

Straub, T. D., Melching, C. S. and Kocher, K. E.: Equations for estimating Clark unit-hydrograph parameters for small rural watersheds in Illinois. No. 2000-4184. US Dept. of the Interior, US Geological Survey; Branch of Information Services, 2000. http://il.water.usgs.gov/pubs/wrir00_4184.pdf

**Table 1.** Simulated input data for initial calibration of the vIUH model for the Hudson River Basin

Hurricane Irene on the Hudson River Basin
Storm duration $\Delta t = 48\ h$
Forecast lead–time 24 $h$

| Station name USGS ID | Basin area $A$ $km^2$ | NARR precip–excess $R_E$ $mm$ | Simulated peak flow $2^{nd}$ $m^3/s$ | $98^{th}$ $m^3/s$ | Mean* $Q(j_p)$ $m^3/s$ | Precip–excess intensity $R_E/\Delta t$ $mm/h$ |
|---|---|---|---|---|---|---|
| Saddle River at Lodi, NJ 1391500 | 141 | 143 | 105 | 200 | 152.5 | 2.979 |
| Hackensack River at New Milford, NJ 1378500 | 293 | 143 | 225 | 442 | 333.5 | 2.979 |
| Walkill River at Gardiner, NY 1371500 | 1800 | 106 | 558 | 1585 | 1071.5 | 2.208 |
| Pomtpon River at Pompton Plains, NJ 1388500 | 329 | 130 | 490 | 1024 | 757.0 | 2.708 |
| Croton River on Hudson, NY 1375000 | 979 | 126 | 503 | 1205 | 854.0 | 2.625 |

Source of data: Saleh et al. (2016).
∗ the average of the peak flows at $2^{nd}$ and $98^{th}$ percentiles.

[Figure]

**Table 2.** Initial calibrated vIUH parameter values for the Hudson River Basin

Hurricane Irene on the Hudson River Basin

$Q(j_p) = 0.2c(R_E/\Delta t)^{1.4} A \Delta t$

$j_p = 0.5 + \frac{0.535}{c(R_E/\Delta t)^{0.4} \Delta t}$

| Station name USGS ID | vIUH parameter $c$ $(mm\ h^{-1})^{0.6}$ | peak time $j_p$ $\Delta t$ |
|---|---|---|
| Saddle River at Lodi, NJ 1391500 | 0.029 | 0.725 |
| Hackensack River at New Milford, NJ 1378500 | 0.026 | 0.711 |
| Walkill River at Gardiner, NY 1371500 | 0.020 | 0.768 |
| Pomtpon River at Pompton Plains, NJ 1388500 | 0.059 | 0.593 |
| Croton River on Hudson, NY 1375000 | 0.024 | 0.728 |

---

## Referee Comment (RC2) · M. Al-Arag (Referee) · 13 Jun 2016

A Retrospective Streamflow Ensemble Forecast for an Extreme Hydrologic Event: a Case Study of Hurricane Irene and on the Hudson River basin

General Comments:

The current manuscript presents the uncertainties in hourly streamflow ensemble forecasts for an extreme horological event using a horological model forced with short-range ensemble weather prediction model. Overall, This is very interesting work and a well-done study. I believe the manuscript is written well.

Specific Comments:

- Streamflow forecasts are indeed highly dependent on the meteorogical input and have historically been associated with much uncertainty.

- Summary/discussion states that a higher confidence in the river discharge forecast may be attained within 48 hours of a major rainfall event. This concept isn't new and points back to the uncertainty with hydrologic modeling.

- In terms of what's new that is presented in this work is perhaps using GIS and a regional scale HEC-HMS model but the paper seems to suggest that you are reducing uncertainty. The work seems to suggest that this will enable better flood forecasts but that can't be ascertained unless you are doing hydraulic modeling (HEC-RAS). It is the hydraulic modeling that will translate the hydrology to a given water surface elevation (i.e., flood stage). It is the flood stage that determines the level of riverine inundation. While HEC-HMS can be used for routing, if this work is suggesting better flood forecasting is possible than traditional riverine hydraulic modeling is warranted. I'm a little unfordable with what is being stated for better control of modeling uncertainty with streamflow forecasts unless this work is coupled with simulations in HEC-RAS. Hydrology is more of science and hydraulics more of an engineering discipline. This work is suitable for hydrologic modeling discussions but not "the uncertainties in weather inputs may result in false warnings and missed river flooding events, reducing the potential to effectively mitigate flood damage" (lines 17-19).

Please also note the supplement to this comment:
http://www.hydrol-earth-syst-sci-discuss.net/hess-2016-104/hess-2016-104-RC2-supplement.pdf

---

## Author Comment (AC2) · 13 Jun 2016

Dear Referee,

Thank you very much for your valuable comments and suggestions that will greatly improve the quality of this manuscript. Indeed, your suggestions will provide a better insight to this paper.

We agree with your comments regarding the concept of obtaining a higher confidence in the river discharge forecast as one approaches the event in question but we would like to reiterate the fact that using ensemble members instead of one deterministic forecast have advantages in terms of better representing the envelope uncertainty,

particularly in an extreme hydrologic event such as the one presented in this work. Furthermore, while the general idea is that hydrologic uncertainty is reduced with lead time, we have not found studies that quantitatively characterize this aspect using the GEFS retrospective data and in the event of an extreme flood event such as Hurricane Irene.

We also agree with you that using a hydrodynamic model will have advantages in terms of simulating the water levels but from our perspective the simulation of hydrological streamflow remains the main driver because HEC-RAS or any other hydrodynamic model would require flow boundary conditions produced by a hydrological (rainfall-runoff) model to solve the Saint-Venant equations. If the flow inputs produced by the hydrological model are not accurate then the simulated inundations or water surface elevations will be impacted regardless of the level of complexity within the modeling framework. Furthermore, hydrodynamic modeling would require detailed representation of river cross sections geometry that is not available at regional scale. In this context, a number of techniques such as pre-defined rating curves (water levels vs. discharge) are operationally used to convert the streamflow to water levels at specific locations, this is also the technique we are currently using in our operational framework.

We will make sure that these issues are appropriately addressed in the discussion part of the paper to satisfy your comments and suggestions.

Respectfully, Firas Saleh

–––––––––––––––––––––––––––––––––

---

## Author Response (AR1)

**Cover letter**

Date: 17-06-2016

To: The Handling Editor of Hydrology and Earth System Sciences

Sub: Submission of revised manuscript hess-2016-104

Dear Prof. Pappenberger,

I would like to first thank you and the reviewers for the insightful comments and suggestions that helped improve the manuscript.

On behalf of my co-authors, I would like to submit the revised version of the manuscript entitled "A Retrospective Streamflow Ensemble Forecast for an Extreme Hydrologic Event: a Case Study of Hurricane Irene and on the Hudson River basin". The manuscript is accepted for publication with minor revisions. We have carefully addressed all the reviewers' constructive comments and suggestions. Along with the revised version, we are also submitting a marked-up manuscript version and a point-by-point reply to the reviewers comments.

Respectfully,

Dr. Firas Saleh

Senior Research Engineer
Stevens Institute of Technology
The Department of Civil, Environmental and Ocean Engineering (CEOE)
Davidson Laboratory
711 Hudson Street, Hoboken, NJ 07030
Tel. 201 216 5654
Email: fsaleh@stevens.edu

**Point-by-point response to the reviews**

**A Retrospective Streamflow Ensemble Forecast for an Extreme Hydrologic Event: a Case Study of Hurricane Irene and on the Hudson River basin**

**Referee #1 comments and reply**

Comment 1: In the abstract the authors state that this modeling framework could be applied anywhere in the world. However, they use NARR dataset and a database from US based gage sites for calibration. How would these methods be applied for watersheds without gaging stations (or with only a few) outside of the US where the NARR dataset does not apply? Also, this statement was not discussed in the paper.
*It is possible to use other sources of atmospheric data in the framework instead of NARR in order to apply the framework to watersheds in other countries. For instance, one may use atmospheric reanalysis products from the European Center for Medium range Weather Forecasting (ECMWF), National Centers for Environmental Prediction (NCEP) and National Center for Atmospheric Research (NCAR). The framework directly handles GRIB1, GRIB2 and NetCDF. As for watersheds without gaging stations (or with only a few), it is possible to use remote sensing river discharge data to calibrate and validate the modeling outputs. Despite that fact that such data have uncertainties, there have been many advancements in this field and there is potential for future applications (e.g., the Surface Water and Ocean Topography (SWOT) satellite mission). This was added in the revised version of the manuscript (section 4).*

Comment 2: The HEC-HMS model uses the SCS Curve Number method that includes "antecedent moisture content" (P5 Line 22) as a parameter for estimating runoff. From my experience, model runoff estimation can be very sensitive to soil moisture. This indicates that calibrating the model will only produce accurate answers for the conditions of the storm it was calibrated to. How do you account for changing soil moisture in the forecast framework?
*This is a very important point that the referee is addressing. We are aware of the limitations in using static parameters for the SCS curve number method. To this end, the framework has a look up table for the initial abstraction parameters based on the hindcast and the continuous run of the model with the NARR data. We are actively working on integrating a machine learning technique which automatically selects the optimal initial abstraction parameters on the fly. This was added in the revised version of the manuscript (section 2.2.2).*

Comment 3: How long does it take for the streamflow forecasts to be produced? How much lead-time is left over? Is it enough to issue a warning?

*For the entire Hudson River Basin, the required time for GEFS (21 members) is around 30 minutes. This includes processing the GRIB files and post-processing of the ensemble outputs. We are currently running 125 ensemble members in the framework and this includes (in addition to GEFS) the ECMWF, ECMWF-HRES, the Short-Range Ensemble Forecast (SREF), the Canadian Meteorological Centre (CMC) and the North American Mesoscale Forecast System (NAM). The total time for all these ensemble members is around five and a half hours from pre-processing to updating the database and the website. We are currently updating the forecasts every 6 hours. The current lead time (87 hours) is sufficient for issuing a flood warning. This was discussed in the revised version of the manuscript (section 4).*

Technical Corrections
*All the technical corrections were carefully addressed in the revised manuscript (see marked-up version).*

**Referee #2 comments and reply**

- Streamflow forecasts are indeed highly dependent on the meteorological input and have historically been associated with much uncertainty.
- Summary/discussion states that a higher confidence in the river discharge forecast may be attained within 48 hours of a major rainfall event. This concept isn't new and points back to the uncertainty with hydrologic modeling.

*We agree with your comments regarding the concept of obtaining a higher confidence in the river discharge forecast as one approaches the event in question but we would like to reiterate the fact that using ensemble members instead of one deterministic forecast has advantages in terms of better representing the envelope uncertainty, particularly in an extreme hydrologic event such as the one presented in this work. Furthermore, while the general idea is that hydrologic uncertainty is reduced with lead time, we have not found studies that quantitatively characterize this aspect using the GEFS retrospective data and in the event of an extreme flood event such as Hurricane Irene. This was added to the revised version of the manuscript (section 4).*

- In terms of what's new that is presented in this work is perhaps using GIS and a regional scale HEC-HMS model but the paper seems to suggest that you are reducing uncertainty. The work seems to suggest that this will enable better flood forecasts but that can't be ascertained unless you are doing hydraulic modeling (HEC-RAS). It is the hydraulic modeling that will translate the hydrology to a given water surface elevation (i.e., flood stage). It is the flood stage that determines the level of riverine inundation. While HEC-HMS can be used for routing, if this work is suggesting better with streamflow forecasts unless this work is coupled with simulations in HEC-RAS. Hydrology is more of science and hydraulics more of an engineering discipline. This work is suitable for hydrologic modeling discussions but not "the uncertainties in weather inputs may result in false warnings and missed river flooding events, reducing the potential to effectively mitigate flood damage" (lines 17-19). Flood forecasting is possible than traditional riverine hydraulic modeling is warranted. I'm a little unfordable with what is being stated for better control of modeling uncertainty

*We agree with the reviewer that using a hydrodynamic model will have advantages in terms of simulating the water levels but from our perspective the simulation of hydrological streamflow remains the main driver because HEC-RAS or any other hydrodynamic model would require flow boundary conditions produced by a hydrological (rainfall-runoff) model to solve the Saint-Venant equations. If the flow inputs produced by the hydrological model are not accurate then the simulated inundations or water surface elevations will be impacted regardless of the level of complexity within the modeling framework. Furthermore, hydrodynamic modeling would require detailed representation of river cross sections geometry that is not available at regional scale. In this context, a number of techniques such as pre-defined rating curves (water levels vs. discharge) are operationally used to convert the streamflow to water levels at specific locations. This is also the technique we are currently using in our operational framework and we have discussed it in the revised version of the manuscript (section 4).*

**Marked-up manuscript version**

[revised manuscript text omitted]

---

## Author Response (AR2)

**Cover letter**

Date: 22-06-2016

To: The Handling Editor of Hydrology and Earth System Sciences

Sub: Submission of revised manuscript hess-2016-104

Dear Prof. Pappenberger,

We have carefully addressed the reviewer comments and suggestions. Along with the revised version, we are also submitting a marked-up manuscript version and a point-by-point reply to his comments.

Thank you for your consideration.

Respectfully,

Dr. Firas Saleh

Senior Research Engineer
Stevens Institute of Technology
The Department of Civil, Environmental and Ocean Engineering (CEOE)
Davidson Laboratory
711 Hudson Street, Hoboken, NJ 07030
Tel. 201 216 5654
Email: fsaleh@stevens.edu

**Point-by-point response to the reviews**

**Reviewer #2 comments and response**

**Your work concerning uncertainty is interesting. Your response seems to suggest that your current work can replace traditional hydraulic modeling/methods and I take exception with that for a couple of reasons:**

**1) Adequate/detailed river cross section geometry is generally available for streams/rivers in NJ/NY and is the case for a well-studied river like the Hudson River. With the advent of LiDAR, river geometry is obtainable and with a greater degree of accuracy then conventional survey methods that have been traditionally used and at a far less expense.**

**For one of your responses you wrote: "hydrodynamic modeling would require detailed representation of river cross section geometry that is not available at regional scale". This detailed representation of river cross section geometry is needed to create an accurate/valid hydraulic model. Your approach reported in this current study is not adequate to replace hydraulic modeling.**

*Thank you for pointing out the relevance of hydraulic modeling to this work. Indeed, it is a very important topic which may potentially expand the scope of this work. The current study was focused on hydrological modeling and associated uncertainties in an operational forecasting context. The current routing technique is based on the Muskingum equations (section 2.2.1) of the revised manuscript. We would like to also emphasize that this work does not suggest to replace traditional hydraulic modeling. In the discussion part of the paper we state "The current work maybe potentially expanded to integrate a hydrodynamic model such as HEC-RAS (Brunner, 2002) that has advantages in terms of simulating the water levels and flood extents in addition to streamflow".*

*Indeed, LiDAR data is becoming more available in certain regions of the world. While there are available datasets at local scales within the NY/NJ areas, we were not able to find readily available data at the scale of the Hudson River basin. Furthermore, LiDAR does not penetrate water bodies so depending on the rivers in question additional bathymetric measurements may be required to complete parts of the river cross sections up to the bed level of the river.*

*We have modified the discussion part to address this comment.*

**2) Indeed boundary conditions (BCs) are needed for hydraulic modeling. BCs can be a rating curve or known water surface elevations (WSEs). USGS maintains (calibrated) gage stations throughout the region and the Hudson River, data is readily available for known WSEs during storms of record in addition to 100-, 500-year discharges.**

*Thank you for discussing this issue. The USGS data are indeed useful for real time boundary conditions while the current work has focused on hydrological forecasts for up to 96-hrs in the future.*

*If a hydraulic model is to be used in future efforts to determine flood extents within a specific forecast horizon then it would require flow obtained from the hydrologic forecasting model. This flow will set the*

*upstream boundary conditions of the hydraulic model in order to solve the one-dimensional unsteady flow equations (momentum and continuity). The USGS observed rating curves may still be used as downstream boundary conditions but not upstream.*

*In this work, the historical streamflow obtained from USGS gage stations of the Hudson River basin were used to calibrate and validate the hydrologic modeling and this was discussed in section 3 of the paper.*

*We would also like to mention that the framework includes data mining tools to automatically retrieve the USGS observed streamflow data in order to force baseflow initial conditions of the hydrological model. This was discussed in section 2.2.2 of the paper.*

**Marked-up manuscript version**

[revised manuscript text omitted]